# Differentiation Disorders of *Chara vulgaris* Spermatids following Treatment with Propyzamide

**DOI:** 10.3390/cells12091268

**Published:** 2023-04-27

**Authors:** Agnieszka Wojtczak

**Affiliations:** Faculty of Biology and Environmental Protection, Department of Cytophysiology, University of Lodz, 141/143 Pomorska, 90-236 Lodz, Poland; agnieszka.wojtczak@biol.uni.lodz.pl

**Keywords:** alga, *Chara vulgaris*, chromatin, electron microscopy, microtubule, microtubular manchette, propyzamide, spermatogenesis, spermiogenesis

## Abstract

Microtubules are cytoskeletal cell elements that also build flagella and cilia. Moreover, these structures participate in spermatogenesis and form a microtubular manchette during spermiogenesis. The present study aims to assess the influence of propyzamide, a microtubule-disrupting agent, on alga *Chara vulgaris* spermatids during their differentiation by means of immunofluorescent and electron microscopy methods. Propyzamide blocks the functioning of the β-tubulin microtubule subunit, which results in the creation of a distorted shape of a sperm nucleus at some stages. Present ultrastructural studies confirm these changes. In nuclei, an altered chromatin arrangement and nuclear envelope fragmentation were observed in the research as a result of incorrect nucleus–cytoplasm transport behavior that disturbed the action of proteolytic enzymes and the chromatin remodeling process. In the cytoplasm, large autolytic vacuoles and the dilated endoplasmic reticulum (ER) system, as well as mitochondria, were revealed in the studies. In some spermatids, the arrangement of microtubules present in the manchette was disturbed and the structure was also fragmented. The observations made in the research at present show that, despite some differences in the manchette between *Chara* and mammals, and probably also in the alga under study, microtubules participate in the intramanchette transport (IMT) process, which is essential during spermatid differentiation. In the present study, the effect of propyzamide on *Chara* spermiogenesis is also presented for the first time; however, the role of microtubule-associated proteins in this process still needs to be elucidated in the literature.

## 1. Introduction

The differentiation of spermatogenic cells is a complex multi-stage process in which, in addition to acetylation, nuclear protein exchange and double-strand breaks of DNA, microtubules also play an important role [1,2,3,4,5,6]. In eukaryotic cells, microtubules form part of the cytoskeleton. The building blocks of these structures, α- and β-tubulin heterodimers, form protofilaments. In plant or animal cells, depending on the stage of their cycle, microtubules are arranged differently and are involved in numerous cell processes, such as cell division, vesicle transport, cellular growth, and the maintenance of a cell’s shape [1,7,8,9]. Microtubules also play an essential role in cell motility and constitute constructive elements of the structures used to move flagella and cilia. To successfully perform these various functions, microtubules are supported by a group of microtubule-associated proteins and posttranslational modifications of tubulin [1,8,10,11,12].

Through the use of different agents, it is possible to influence the degree of polymerization and depolymerization of microtubules, and these processes are used in the fields of agriculture [13] and medicine to treat numerous diseases, such as cancer [14]. The majority of the research conducted on the process of microtubule assembly, as in the case of other issues analyzed in the research, is mostly focused on animals [6,15,16,17], some plants, and, to a small extent, algae [12,18,19,20,21].

Propyzamide (also called pronamide), a compound binding to β-tubulin, is one of the agents that destabilizes microtubules [22]. Several studies are concerned with the impact on the ecosystem of different inhibitors used individually or in mixtures, including the impact of propyzamide on algae (Chlorophyta and Rhodophyta) [23,24,25,26], plants, invertebrates, (*Daphnia* sp.) and vertebrates (fish and frogs) [27,28].

Plant cell studies include analyses of the effects of various agents on the cell cycle and cytoskeleton, as well as on the plant organ growth disorders related to them [29,30,31,32]. In the case of algae, studies have been conducted with the use of several species, including the unicellular alga *Chlorella vulgaris* [33], and also concerned the participation process of microalgae during bioremediation [24,25,34]. The research performed on algae of the Charophyceae class concerns cortical microtubules and their depolymerization behavior [18,19,35,36]. These studies are mostly conducted on the large, internodal thallus cells of *Nitella* and *Chara*, which provide a convenient model for the process of cytoplasmic streaming [36,37,38,39]. The effects of various agents, dinitroaniline and propyzamide, on cortical microtubules in the rhizoids of the alga *Spirogyra* [40], and the effects of cytochalasin and propyzamide on cytoplasmic streaming in the root hair cells of *Hydrocharis dubia* have also been investigated [37]. The research conducted on the effect of propyzamide on cell divisions was conducted on the species of Rhodophyta (*Cyanidioschyzon merolae*) [23] and the *Cyanophora sudae* unicellular alga [41].

In *Chara*, during spermatogenesis, following the proliferative phases, antheridial filament cells enter the differentiation phase (spermiogenesis) and transform from somatic into generative cells. It is a complicated and long-term process of which ten stages (I–X) characterized by specific features have been distinguished in the literature [42]. During *Chara vulgaris* spermatid differentiation, structures built out of microtubules, i.e., the microtubular manchette in the cytoplasm and two flagella between the plasmalemma and cell walls of antheridial filaments, are formed [43]. The microtubular manchette has also been discovered in spermatids of certain groups of animals, such as mammals [44,45], birds (poultry [46] and ostriches [47]), reptiles [48,49], and octopuses [50]. However, the process of manchette formation and the period of time for which it is present differ [6,43,51,52,53]. In the case of *Chara*, the microtubular manchette [42,43] is formed during the transitional period between stages IV and V and persists until the end of spermiogenesis, while in the above-mentioned groups of animals it is a temporary structure. As a result of these differences, it is sometimes difficult to analyze and compare the effects of microtubules disrupting inhibitors on spermiogenesis. In *C. vulgaris*, the manchette is formed by linearly arranged microtubules located under the plasmalemma. Complexes of microtubules and proteins connected to them participate in the intramanchette transport (IMT) process, which ensures the proper course of spermiogenesis [5,52,54]. One of them is the RIM-BP3 protein, which is bound to the microtubules of the manchette, and it is essential for the spermatogenesis that occurs in mice [16].

In the literature, at present, the influence of various factors that disrupt the proper functioning of microtubules, especially in relation to alga spermiogenesis, is not extensively analyzed [18,19,35]. Moreover, the effect of propyzamide on *Chara* spermatogenesis has not been analyzed to date. However, the influence of another compound, dinitroaniline, on the elements of the cytoskeleton during spermatogenesis has been investigated on *Chara contraria*, but not on the ultrastructural level [18].

The aim of the present study is to analyze the impact of propyzamide on *C. vulgaris* spermatids, during their differentiation. It is well-known in the literature that the microtubular manchette contributes to maintaining the shape of the nucleus of spermatids and participates in IMT. Since propyzamide binds to β-tubulin and destabilizes microtubules, the analysis of β-tubulin immunolabeling and the ultrastructure of maturing spermatids permitted the assessment of how this compound affects the functioning of these cellular structures. In the present paper, the influence of this microtubule function disrupter on the process of *C. vulgaris* spermiogenesis is presented for the first time.

## 2. Materials and Methods

### 2.1. Material

The analysis performed in this study concerned *Chara vulgaris* (Charophyceae) antheridia derived from the pleuridia of the III–V nodes. The sampled algae were grown in and collected from an artificial pond in the Rogów Arboretum (Warsaw University of Life Science, Warszawa, Poland), cultivated at room temperature, in accordance with the method of Wojtczak and Kwiatkowska [55], and then used for the present research. Spermatids at all spermiogenesis stages (I–X) were analyzed.

### 2.2. Treatment with Propyzamide

The thallus fragments carrying antheridia were incubated for 48 h in propyzamide, a microtubule-destabilizing factor (Sigma 45645), at a concentration of 50 µM. Water collected from the natural environment and dimethyl sulfoxide (DMSO) (Sigma D-4540) in a 0.1% ratio, administered by Akashi et al. [30], were used to prepare the propyzamide solution. The control material was treated only with DMSO in the same ratio as that of the propyzamide solution. The solutions were then prepared with water collected from the natural environment in order to not affect the conditions for the algae’s growth.

### 2.3. Immunocytochemical Studies of β-Tubulin

The isolated antheridia of *C. vulgaris* were fixed in 10% formalin and embedded in paraffin in accordance with the procedure presented in the study conducted by Wojtczak [56]. Following the initial preparation of the sections, including the performance of deparaffinization with xylene, gradual hydration using a series of alcohols, macerozyme treatment, and permeabilization with 0.1% Triton X-100, the preparations were incubated overnight at 4 °C with a primary antibody (mouse monoclonal antibody raised against β-tubulin, Sigma T5201) diluted at 1:300 in phosphate-buffered saline (PBS) containing 5% bovine serum albumin (BSA) and 0.5% Tween 20, washed with 0.2% Tween 20 in PBS. Then, the appropriate sections were incubated with the secondary antibody (antimouse IgG conjugated with FITC; Sigma F1262) diluted at 1:70 in PBS containing 5% BSA and 0.5% Tween 20 for 2 h and stained using DAPI in darkness at room temperature. Following the staining procedure, the preparations were embedded in a PBS/glycerol mixture (9:1) with 2.3% DABCO (1,4-diazabicyclo-[2,2,2] octane, Sigma). The manner of taking the photos was consistent with that presented in the study conducted by Wojtczak [56]. Negative control sections were not treated with the primary antibodies in the experiment.

The fluorescence intensity in spermatids was measured from selected spermiogenesis stages (presented in Figure 1) for the control and propyzamide-treated variants, using ImageJ (Appendix A). The average value and standard deviation (means ± SD) were calculated in Microsoft Excel 2000.

### 2.4. Electron Microscopy

Specimens of *C. vulgaris* antheridia collected from the control sample and the material treated with propyzamide underwent the standard electron microscopy embedding procedures outlined in the previous paper [3]. The material was fixed with 3% glutaraldehyde in a 0.1 M cacodylate buffer (pH 7.3) supplemented with 0.007 M CaCl2 for a period of 3 h. The material embedded in the Epon 812 and Spurr mixture medium (Polysciences) was cut and applied to electron microscopy grids. An additional standard procedure performed for contrasting ultrathin sections was conducted in accordance with the procedure of Reynolds [57]. The ultrathin sections were then examined and photographed using a JEOL JEM-1010 transmission electron microscope (TEM).

## 3. Results

### 3.1. Immunocytochemical Studies

The analysis of antheridial filaments obtained from the control sample and material treated with propyzamide revealed differences in some spermatids’ nuclei and in the localization of β-tubulin immunosignals (Figure 1).

In the control group, during the initial stages of spermiogenesis (I–III), weak immunosignals were measured (Figure 1(I–IIIA) and Appendix A). Following propyzamide treatment, single, very weak or no immunosignals were visible (Figure 1(I–IIIC)). During stages IV–V, the immunoreaction that occurred was much stronger in the control sample (Figure 1(IVA,VA)) than it was after the treatment performed with propyzamide (Figure 1(IVC,VC)). In the control sample, signals in the cytoplasm located on both sides of the spermatid’s nucleus were evident (stage IV, Figure 1(IVA)) or they formed clusters aligned in a bowlike shape along the edge of the cytoplasm (stage V, Figure 1(VA)). At stage IV, in the treated spermatids, some very weak, single immunosignals were visible (Figure 1(IVC)), while at stage V, they were stronger than during the previous stage; however, they were arranged at longer distances along the cytoplasm area (Figure 1(VC)) in comparison to the control sample (Figure 1(VA)). During the later stages of spermiogenesis (VI–VIII), in the control sample, numerous distinct foci were present on the outer perimeter of the cytoplasm. They were clustered and arranged in the shape of an arc (Figure 1(VI–VIIIA)). During the same stages (VI–VIII), following propyzamide treatment, only at stage VI could single, pale foci be observed on the cytoplasm (Figure 1(VIC)), and at stages VII and VIII, no immunosignals were visible (Figure 1(VIIC,VIIIC)). At the late stages of spermiogenesis (IX–X), in the control sample, the immunoreaction was weaker than previously observed. The immunosignals were present in the thin layer of the cytoplasm surrounding the spirally folded spermatozoid (Figure 1(XA)). At the final differentiation stages, following propyzamide treatment, there remained no evidence of fluorescent foci (Figure 1(XC)).

The analysis of DNA staining using DAPI presented the distorted shape of the nucleus in some spermatids treated with propyzamide (stages V, VII, and VIII) (Appendix A) in contrast to the control images (Figure 1). The spermatid nuclei were fuzzy and did not present distinct borders. The chromatin seemed to be diffused and its area was not limited by the shape of the nucleus. The sections in which the antibody to β-tubulin was omitted (negative control) were free of immunoreaction (not presented). A summary of the effects of β-tubulin immunoreaction are presented in Table 1.

### 3.2. Ultrastructural Studies

The analysis of antheridial filament fragments collected from the material treated with propyzamide presented a diversity of ultrastructural changes during the spermatid differentiation process in comparison to the control sample (Figure 2). The most important effects of propyzamide, separately for the nucleus and cytoplasm, are presented in Table 1.

The ultrastructural changes were visible both in the nucleus and cytoplasm areas; however, they were not observed in the entire cell area in all spermatids at a specific stage of spermiogenesis. Sometimes, the propyzamide produced strong changes in the ultrastructure of the spermatids so it was difficult to precisely determine the stage of spermiogenesis. In some antheridial filaments (stages IV and V), the disturbances were so strong that they also occurred in the cell wall, which was somewhat stratified and less visible (Figure 2(III–IVD,VC)).

Changes occurring in the nucleus and cytoplasm of the spermatids are presented separately in the Results section. The results concerning stages I–V, VI–VIII, and IX–X of spermiogenesis are also discussed in detail. Stages I–V are discussed together because of the similarities concerning the changes produced by the propyzamide and arrangement of the condensed chromatin, adjacent to the nuclear envelope, and the non-condensed chromatin, located in the central part of the nucleus.

#### 3.2.1. Nucleus

At the early stages of spermiogenesis (I–IV), in some spermatids treated with propyzamide, changes in the nucleus were either invisible or visible to a greater or lesser extent (Figure 2(III–IVB),C). It can be observed that the effect of propyzamide on these spermatids was more pronounced in the cytoplasm (Figure 2(III–IVC)).

In some early formed spermatids, the change in the nucleus concerned only the nuclear envelope that, in contrast to the control sample (Figure 2(III–IVA) and Figure 3A), had numerous shallower or deeper invaginations. These images were observed at stages IV (Figure 3B) and V (Figure 3C,D), during which the nuclei were characterized by condensed chromatin near the nuclear envelope. In the case of spermatids with very strong nuclear structural disturbances, the fragmented nuclear envelope was visible (Figure 2(III–IVD,VC) and Appendix A).

The sample took the form of either shorter or longer fragments, which were often much wider, as if they were inflated (Appendix A). The nucleus shape following propyzamide treatment remained mostly unchanged (Figure 2(VB)) compared to the control sample (Figure 2(VA)). Despite the invagination and fragmentation of the nuclear envelope, the area occupied by the nucleus was easily recognizable (Figure 2(III–IV,V), Figure 3C and Appendix A). Due to the alteration of the spermatid’s structure induced by the propyzamide, it was not always possible to accurately determine the stage of spermiogenesis by only observing the location and structure of the nucleus.

In most spermatids at stages III–V following propyzamide treatment (Figure 2(III–IVB,C,VB)), the arrangements of condensed and non-condensed chromatin were similar to those of the control sample (Figure 2(III–IVA,VA) and Appendix A). The condensed chromatin was located near the nuclear envelope and the non-condensed chromatin was located in the remaining part of the nucleus. However, a less compact system of non-condensed chromatin fibrils was present (Appendix A) in the nuclei (stages III–IV), despite the correct distribution of both types of chromatin. Sometimes, the condensed chromatin lacked the presentation of a typical structure and resembled homogeneous spots of electron-dense materials. On the circumference of the condensed chromatin, there were numerous long gaps devoid of it and of the nuclear envelope (Appendix A). The condensed chromatin also formed regularly shaped spots in the central section of the nucleus (Appendix A). In the non-condensed chromatin region, there were numerous chromatin-free sites present, either in the central section of the nucleus or in the immediate vicinity of the condensed chromatin (Appendix A). Some nuclei presented an openwork structure, in which there were areas with spaces in the center. There were also areas devoid of chromatin fibrils and it was difficult to recognize the non-condensed chromatin (Appendix A). In some nuclei of the spermatids, an extensive nuclear reticulum (NR) system was observed (Figure 3B and Appendix A) much earlier on in the process than in the control sample, in which it was present at stage V.

At stages VI–VIII in the nucleus following propyzamide treatment, chromatin condensation and arrangement processes were clearly disturbed. The condensed chromatin took the form of streaks or spots that, in some spermatids, were thicker, and more closely packed and in others were arranged as thin, irregularly shaped bands (Figure 2(VIB,VIIIB), Figure 4A,B and Appendix A). At stage VI when, in the control sample, the condensed chromatin formed a characteristic network (Figure 2(VIA)), following propyzamide treatment, its filaments were visible; however, they did not form the network system, and between them there were large, brighter electron spaces (Figure 2(VIB) and Appendix A). At stages VII and VIII, in some propyzamide-treated spermatids, the chromatin arrangement was altered. The condensing chromatin fibrils were dispersed across the nucleus (Figure 4A,B), instead of being arranged in parallel, similarly to the control sample (stage VIII, Figure 2(VIIIA)). Moreover, in some images, it was also difficult to observe a clear boundary between the nucleus and cytoplasm (Figure 4A,B) because the shape of the former was changed.

In the late-stage spermatids (IX, X) and, similarly, at stages VI–VIII, the process of chromatin condensation following propyzamide was incorrect. In the control sample, the nuclei became elongated and adopted a spiral shape (two coils at stage X), and numerous transverse sections of the nucleus at certain distances, near both sides of the cell walls of the antheridial filaments (Figure 2(IX–XB)), became visible. However, in some spermatids, following treatment with propyzamide, an arrangement of multiple nucleus transverse sections, often lying next to each other, was present (Figure 2(IX–XC) and Figure 4D,F). In some sections, the chromatin presented a homogeneous or fibrillar structure with bright electron spaces in between (Figure 4D–F,H), and often formed a patchwork pattern (Figure 4C,G). Despite the abovementioned disorders, there was also a presence of nuclei with correctly condensed chromatin (Figure 4I).

#### 3.2.2. Cytoplasm

The cytoplasm of treated spermatids was characterized by a much looser structure. In the vast majority of spermatids in the early stages (I–IV) of spermiogenesis, the changes induced by propyzamide mainly affected the cytoplasm. Additionally, numerous spaces, smaller or larger, empty or with contents (Figure 5C–J), were observed.

Moreover, many endoplasmic reticulum (ER) cisternae were visible (Figure 5D,E,G); in the control group, their number was very high, only at stage V (Figure 2(VA)). ER cisternae were filled with bright-electron material or granularities. A system of dilated ER cisternae or vesicles with light-gray contents was also observed. In the cytoplasm, numerous Golgi apparatuses, membranous structures, and small vesicles, probably autolytic vacuoles, were also visible in the images (Figure 5D,E,G–J). In some spermatids, mitochondria with distended cristae (Figure 5D,G–I), in contrast to the control spermatids (Figure 5A), were present. At stages IV–V following propyzamide treatment (Figure 5K and Appendix A), similarly to the control sample (Figure 5B), a microtubule manchette, characteristic of spermatids, was visible. The microtubules were arranged linearly under the plasmalemma. In the propyzamide-treated spermatids, microtubules forming the manchette and flagella did not present clear differences in contrast to the control sample.

However, the microtubular manchette was discontinuous and formed several shorter or longer fragments near the plasmalemma (Figure 5K and Appendix A) in the spermatids at these stages (IV–V), where significant disturbances in the chromatin and cell wall (Appendix A–H) were observed in the experiment. There were numerous images in which two to three microtubules were positioned more loosely, with larger spaces between them (Figure 5K and Appendix A), in comparison to the control image (Figure 5B). These fragments, probably devoid of microtubules, appeared alone or in close proximity to the normal manchette fragments. At the same stages, the cytoplasm was more loosely arranged and somewhat diffused (Appendix A). A dilated ER system was also observed (Appendix A).

In the control sample, during the middle stages of spermiogenesis (stages VI–VIII), similarly to the earlier stages, there was evidence of a gradual reduction in the quantity of the cytoplasm present. Following propyzamide treatment, the cytoplasm was not always reduced as it was in the control sample; however, it still covered a significant area in the spermatids. In some cases, the boundary between the nucleus and cytoplasm was not clearly visible (Figure 4A,B and Figure 6A). Numerous ER vesicles were evident, the fine-grained content of which was similar to that observed in the perinuclear space and autolytic vacuoles during earlier stages (Figure 6A,B and Appendix A). On the spermatid cross-sections, fragments of the manchette composed of several microtubules near the nuclei and mitochondria with swollen cristae were visible (Figure 6A,B).

At the end of the spermiogenesis process (stages IX and X), cytoplasm still remained abundant in the spermatids collected from the propyzamide-treated material (Figure 2(IX–XC) and Figure 6D,F–J), compared to the control sample (Figure 2(IX–XA,B)). The cytoplasm was either electron-homogeneous or its structure resembled images of the earlier stages in the process (VI–VIII); however, many larger autolytic vacuoles, filled with granular or membranous material, were present in the central section or in the periphery of the spermatids (Figure 6D,F,H–J). Despite serious disturbances occurring in the cytoplasm and nucleus, in some spermatids, fragments of the correct microtubular manchette remained present between the nucleus and plasmalemma (Figure 6C,E,F–J).

As previously mentioned, in *C. vulgaris* spermatids, microtubules form two flagella. TEM images of the propyzamide-treated material showed that they developed properly, without presenting noticeable irregularities; numerous cross-sections of these structures were visible (Figure 2, Figure 4D,G, Figure 6B,D and Appendix A), similarly to the control sample. On the transverse section of the flagellum, the arrangement of microtubules was also correct (Figure 6C,E and Appendix A).

In the control sample in the antheridial filaments, spermatids persisted in the central section of the cell (Figure 2), whereas following propyzamide treatment, changes in their orientation were observed at stages VIII–X (Figure 2(VIIIB,IX–XC) and Figure 4D). Some of these spermatids were not always arranged parallelly to the transverse cell walls of the antheridial filaments, as they were in the control sample, and, additionally, they were shifted to one of the side walls. The cross-sections of the nuclei in the treated and control materials during stages IX–X revealed tighter spiral coils in the former material (Figure 2(IX–XC) and Figure 4D).

## 4. Discussion

The spermatogenesis of *Chara*, the model organism, has been studied for numerous years in the field [42,43,55]. As mentioned in the Introduction, algae are not a group of organisms that have been specifically studied, as far as the effects of different substances on their spermiogenesis are concerned. The research conducted on the participation of microtubules in spermatogenesis mostly considered animal material, mainly mammals [2,5,58], and the role of the cytoskeleton during this process is still examined in the literature.

During the analysis performed on the spermiogenesis process, it should also be noted that, although these processes occurring in mammals and *Chara* present numerous similarities [3,4,55,56,59], sperm develops in different cell systems. In the literature, there remains little information showing the extent to which tubulin function disturbances caused by different inhibitors can affect cells during the process of spermatogenesis [5,18,19,58,60,61].

The immunocytochemical and ultrastructural studies conducted at present reveal that the interaction of propyzamide with tubulin leads to extensive changes in whole spermatids.

### 4.1. Immunocytochemical Studies

Studies conducted on human spermatogenic cells show that the presence of tubulin and DNA strand breaks might be determinants of the changes observed in incorrect spermatozoa [62]. The abnormalities evident in the development of the microtubular manchette were also revealed in the antitubulin immunofluorescence analysis of mouse spermatids exposed to vinblastine [58]. Similarly, in *C. vulgaris,* immunofluorescent studies of β-tubulin showed that, in the treated material, immunosignals only occurred at stages IV–VI. The low intensity of these immunosignals (stage VI) or their absence in the later stages (VII–X) was caused by blocked-functioning β-tubulin subunits. In the control sample, at the late stages of the spermiogenesis of the studied alga, a β-tubulin immunoreaction was very poorly visible, similarly to microtubule staining during the cell differentiation process of *C. contraria* [18], during which, in spermatozoids, the presence of microtubules was only revealed in the flagella, but not in the manchette. In *C. vulgaris* treated with propyzamide, similarly to *C. contraria* treated with dinitroaniline [18], in some spermatids, microtubules of the manchette were discontinuous or reduced and the nuclei presented an altered chromatin structure.

### 4.2. Ultrastructural Studies

These immunofluorescent observations regarding the altered arrangement of the manchette microtubules and diffused chromatin following DAPI staining analysis were also confirmed in ultrastructural studies. The invaginations of the nuclear envelope in *C. vulgaris* resembled an electron micrograph of the root cells of *Triticum aestivum* treated with another type of inhibitor [63]. In the nucleus, cytoplasmic invaginations and numerous vesicles were visible; they were similar to the NR present in *Chara*. Similarly, an alteration in the nucleus shape caused by nuclear lamina invaginations was also observed in Brg1 knock-down mammary epithelial cells [64]. It can be assumed that nuclear envelope invaginations occurring in the studied alga were also the effect of the disruption of the SWI/SNF chromatin remodeling complex. The Brg1 protein, a subunit of this complex, was abundant at stage IV [4], in which these changes, following propyzamide treatment, were observed. In addition to nuclear lamins, disturbances in nucleoporin expression also led to various changes in the shape of the nucleus [9].

Studies conducted on several inhibitors have shown that they affect the membrane’s ultrastructure. Ruptured and degraded envelope membranes were observed in the chloroplasts and mitochondria [65]. Similar abnormalities in the structure of the nuclear envelope were observed in the spermatids of the studied alga treated with propyzamide and, to some extent, they resembled the results attained for mouse mutants in late spermatids (stage 11) [66]. Chromatin disturbances observed in *C. vulgaris* spermiogenesis were also probably associated with nuclear envelope fragmentation occurring in some spermatids at stages IV and V. At these stages, the cytoplasm was somewhat diffused because the plasmalemma was presumably partially damaged. The proper functioning of the nuclear envelope is necessary in the process of ribonucleoprotein migration into the cytoplasm and during protamine synthesis [42]. The multispectroscopic studies conducted on calf thymus DNA showed that propyzamide changed the DNA conformation form [67]. It can be assumed that the changes observed in the nuclei of the spermatids of the studied alga may also partially be the result of similar damage to DNA.

Ultrastructural changes induced by propyzamide concerned, e.g., the ER system and manchette, and resembled the defense mechanism against the harmful effects of propyzamide. The disturbances in the ER system caused as a result of ER stress were also observed in different cell types and also in *C. vulgaris* following treatment with bromodomain inhibitor JQ1 [59,68,69]. In the internodal cells of *Nitella* (Characean) following dinitroaniline treatment, the cortical ER changed its orientation while increasing the size of the mesh [70]. In the case of *C. vulgaris* spermatids, following propyzamide treatment, extensive cisternae and vesicles in the ER system probably partially occurred as the result of the fragmentation of these cisternae. The microtubular manchette is a characteristic structure in differentiating spermatids. In propyzamide-treated *C. vulgaris* spermatids, the configuration of microtubules forming the microtubular manchette resembled that in the control sample or was of a loose arrangement in fragments. Probably, similarly to the epidermis of the *O. umbellatum* ovary following propyzamide treatment [71], despite the presence of microtubules and their proper appearance, their function was disturbed, as presented by the change in the nucleus’ shape following DAPI staining and by ultrastructural changes in the spermatids. It should be noted that the microtubules in the manchette and flagella of spermatids are more stable compared to cortical and spindle microtubules; therefore, the effects of propyzamide and other inhibitors may vary depending on the type of cell and even their application time [2,18,29,30,31].

Complexes of microtubules and the proteins connected to them participate in IMT, which ensures the proper course of spermiogenesis [5,52,54]. Numerous studies conducted on mammal spermatogenesis have been conducted in the literature [44,52,53,66,72]; however, the issues related to the involvement of microtubules in IMT during spermiogenesis have not been explained [52,54]. In addition, since the 26S proteasome also participates in IMT [52,54], the disruption of proteolytic enzyme delivery affects the abnormal exchange of nuclear proteins, observed as a change occurring in the arrangement of chromatins. Moreover, during mouse spermatogenesis, an important role of intraflagellar transport (IFT) has also been demonstrated in the literature, in which the DLEC1 protein that interacts with tubulin subunits is crucially involved in the process. A deletion of the *Dlec1* gene disrupts the functioning of the microtubular manchette and male fertility [73]. In the case of *Chara* spermiogenesis, no detailed analyses of the functions performed by IMT exist in the literature. At the ultrastructural level, it is difficult to compare the present results with those attained for spermiogenesis studies conducted on other algae. It is well known in the literature that, in mammals, the microtubular manchette and related IMT play a significant role in spermatid chromatin remodeling, sperm function, and maintaining the correct shape of the sperm head [5,44,54].

On the basis of the previous studies conducted on *C. vulgaris* spermatids, it is well known that the inhibition of the activity of the 26S proteasomes, DNA topoisomerase II, and bromodomains, which are necessary for the correct exchange of histones into protamines, causes disturbances in the structure of chromatin; however, the nucleus shape and microtubular manchette were not altered [3,55,59]. Only in the case of treatment with JQ1, following Feulgen staining, some nuclei at stage X were thinner than that of the control sample, presenting an altered surface [59]. It seems likely that the observed changes in the chromatin structure in the present study were the effect of a disturbance in IMT related to the different arrangement or loss of microtubules in the manchette and the fragmentation of the nuclear envelope, which plays an important role in the process of nucleus–cytoplasm transport.

During the early stage of mouse spermiogenesis, following nocodazole treatment, which interferes with microtubule polymerization, cortical microtubules are disassembled, and the Golgi apparatus and acrosome are fragmented [72]. In *C. vulgaris*, the most severe disturbances of the microtubule system following propyzamide treatment were visible when the manchette was formed (at stages IV–V). In the elongated spermatids of these species, the microtubules forming the manchette were less sensitive to the respective treatments [72], present studies.

At some stages, the cell wall in antheridia filaments in *C. vulgaris* following propyzamide treatment was less visible, as if it was stratified. One can only assume that this cell wall image was the result of the fact that, during these stages, cortical microtubules, which can probably participate in the processes of cell wall regeneration (as shown for alga *Mougeotia* [74]), disappeared (as shown for *C. contraria* [18,19]). In another unicellular green alga sample, *Penium margaritaceum*, under the influence of dinitroaniline, a thinner cell wall was evident [75].

The formation of the correct gametes is crucial for the adequate reproduction of organisms. Disorders concerning the microtubular manchette, proteins associated with them, flagella, and head shape, contribute to the formation of abnormal spermatids [5,54,58,61,66]. Studies conducted on mice and *Chlamydomonas* showed that the disruption of a good axoneme structure contributed to the disruption of sperm motility [76]. Different chemicals were also shown to be detrimental to animal and human reproduction processes [69,77,78,79]. All of these abnormalities can ultimately lead to male infertility [79,80], which is a serious problem, particularly for human beings. Studies conducted on mouse spermatogenesis using a chemotherapeutic agent, vincristine, which inhibits the functioning of spindle microtubules, are important for cancer treatment regarding testicular tissue in human beings [81,82]. Kumamonamide, a new inhibitor isolated from *Streptomyces werraensis*, disturbs the functioning of microtubules in several plant species; however, it is safe for the growth of HeLa and *Escherichia coli* cells [20]. Recently, a tubulin inhibitor, Todalam, with a specific mechanism of action that inhibits mammalian cell growth, was designed [83]. Research on the participation of microtubule-associated proteins in the regulation of microtubule dynamics and in plant development is being conducted, at present. Furthermore, it has been shown in the research that the MICROTUBULE ORGANIZATION 1 (MOR1) protein is essential to mitosis and cytokinesis processes in *Arabidopsis*. A seedling analysis showed that MOR1 mutants are more sensitive to propyzamide treatment [84].

In the case of *Chara*, it was difficult to identify possible motility disorders and/or the impact on subsequent generations of spermatids, since, in human beings and alga, meiosis occurs during different stages of their life cycles. However, it can be stated that, similarly to those in mammals, the manchette’s microtubules played an important role during the spermiogenesis of the studied alga. The disruption of proper microtubule functioning contributed to extensive changes in the differentiating spermatids. This issue requires further research, and additional studies on this topic are planned for the future.

## Figures and Tables

**Figure 1 cells-12-01268-f001:**
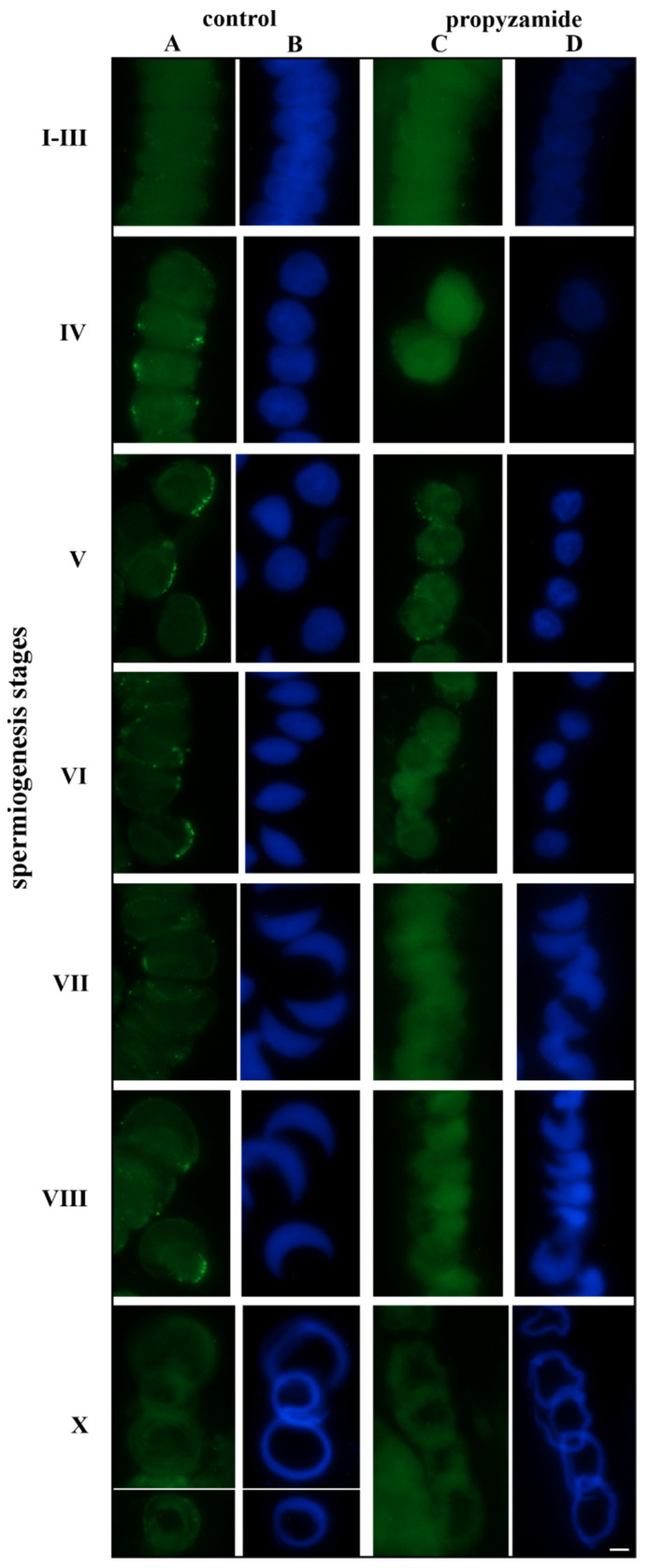
Immunocytochemical localization of the β-tubulin protein in antheridial filament cells obtained from the selected stages of *C. vulgaris* spermiogenesis. (**A**,**B**) Control stages: clear reaction—stages IV–VIII, (**C**,**D**) following 48 h treatment with propyzamide; weaker reaction—stages IV–VI, with secondary antibodies conjugated with FITC (**A**,**C**); positive fluorescence is marked with green foci. Nuclei stained with DAPI (**B**,**D**) are colored in blue. Scale bar: 10 µm.

**Figure 2 cells-12-01268-f002:**
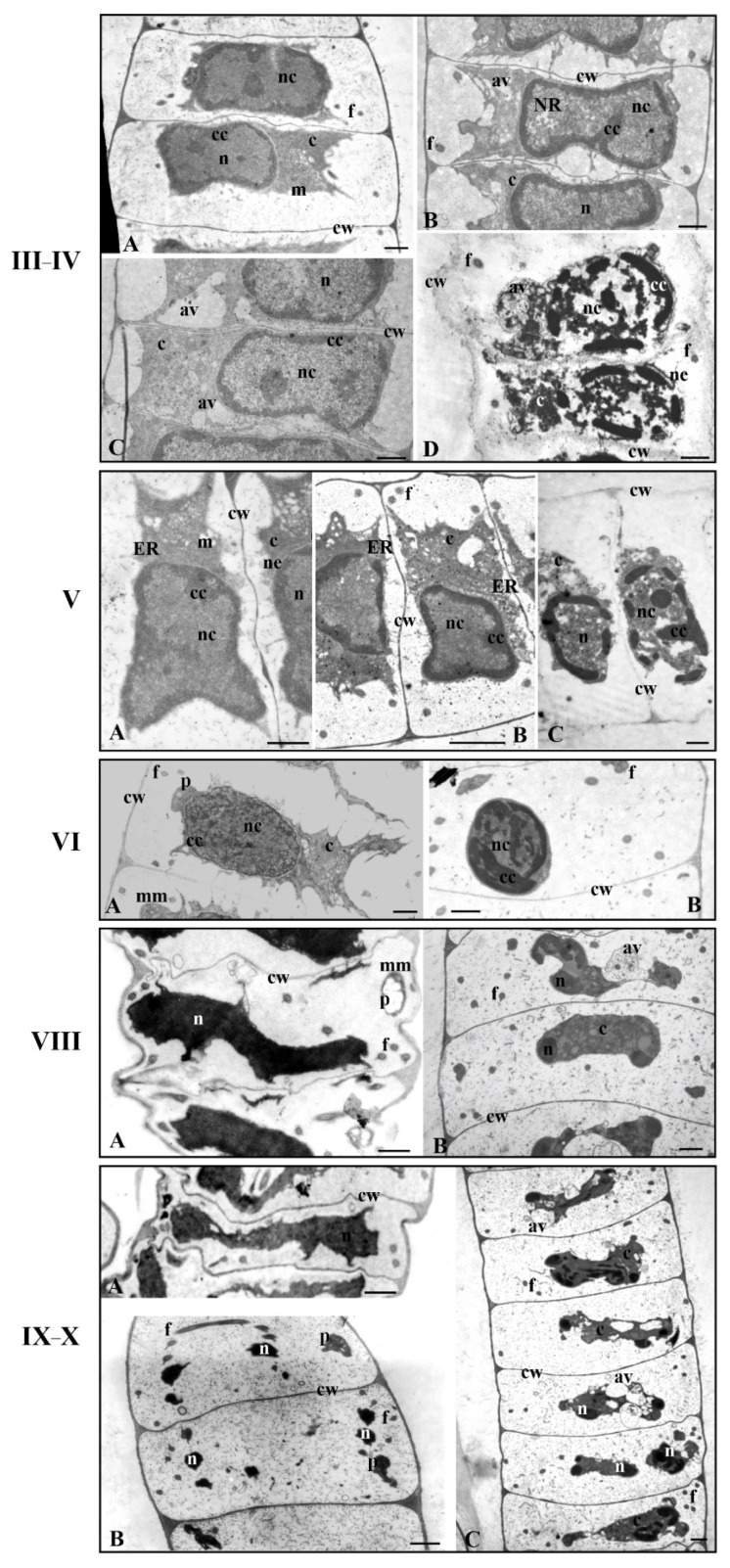
Longitudinal sections of antheridial filament fragments with ultrastructural changes following treatment with propyzamide. Disturbances occurring in both chromatin condensation and cytoplasm areas. Spermatids obtained from the selected stages (III–X) of C. vulgaris spermiogenesis. Control spermatids—(**A**) at all presented stages and (**B**) at IX–X stages; spermatids following propyzamide treatment—(**B**) at stages III–IV, V, VI, VIII, (**C**) III–IV, V, IX–X and (**D**) III–IV; av, autolytic vacuole; c, cytoplasm; cc, condensed chromatin; cw, cell wall; ER, endoplasmic reticulum; f, flagellum; mm, microtubular manchette; n, nucleus; nc, non-condensed chromatin; ne, nuclear envelope; p, plastid. Scale bar: 1 µm.

**Figure 3 cells-12-01268-f003:**
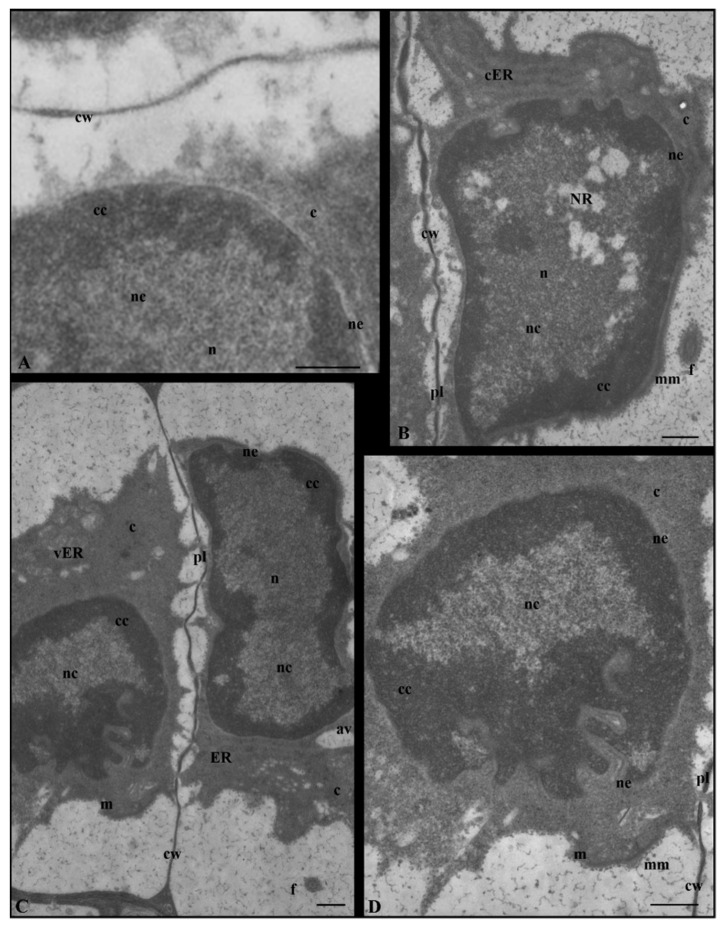
The invaginations of the nuclear envelope of propyzamide-treated spermatids of *C. vulgaris* at stages IV and V. (**A**) Control stage IV. (**B**) Extensive system of nuclear reticulum (NR) in non-condensed chromatin area at stage IV; numerous nuclear envelope invaginations. (**C**) Two spermatids, at stages IV (right) and V (left). (**D**) Higher magnification of the spermatid at stage V shown in (**C**); deep invagination, unchanged chromatin arrangement. Longitudinal section of spermatids; av, autolytic vacuole; c, cytoplasm; cc, condensed chromatin; cER, endoplasmic reticulum cisternae; cw, cell wall; f, flagellum; m, mitochondrion; mm, microtubular manchette; n, nucleus; nc, non-condensed chromatin; ne, nuclear envelope; NR, nuclear reticulum; pl, plasmodesmata; vER, endoplasmic reticulum vesicle. Scale bar: 500 nm.

**Figure 4 cells-12-01268-f004:**
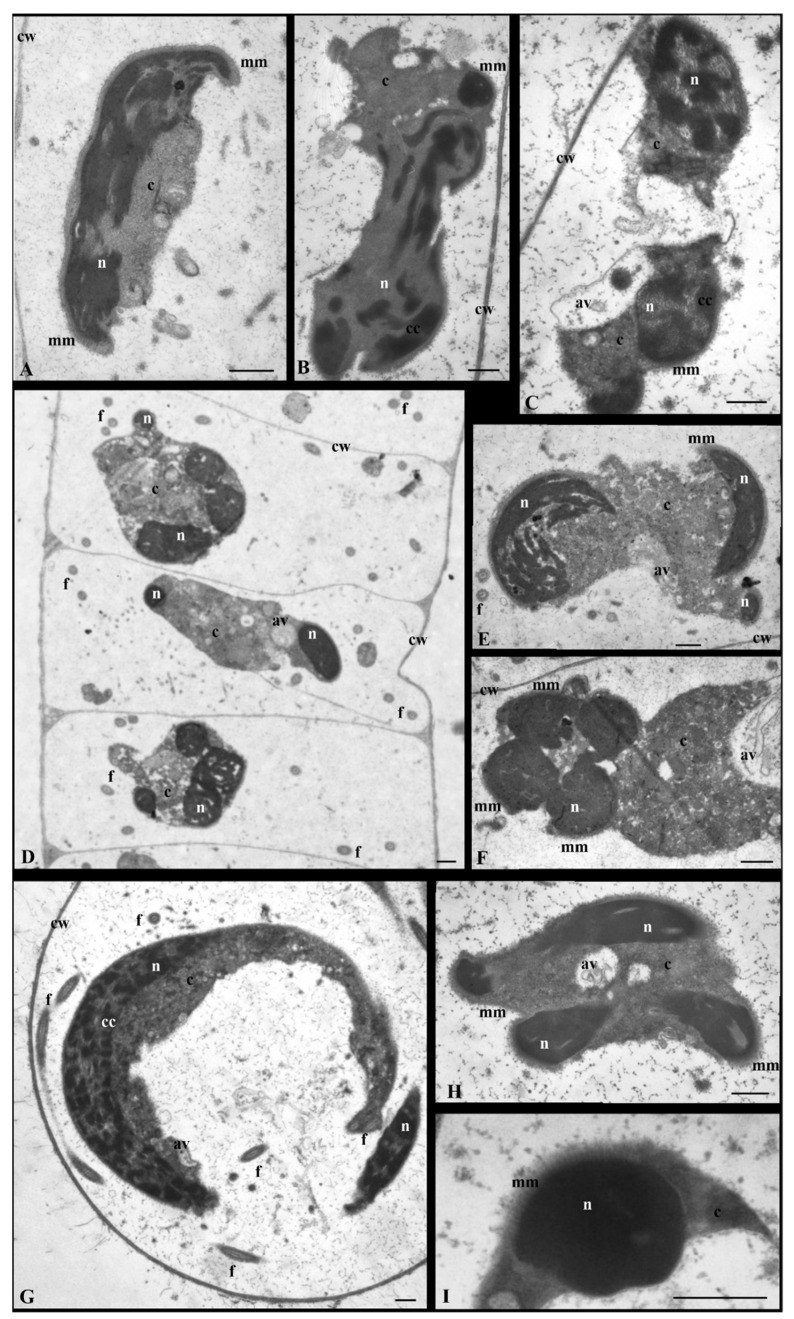
Ultrastructural changes in the spermatid nuclei treated with propyzamide at stages VII–X. Disturbances in the chromatin arrangement. The condensed chromatin dispersed in the nucleus area. In some spermatids, multiple nucleus transverse sections are often positioned next to each other. (**A**,**B**) Chromatin in the form of homogenous bands. (**C**,**G**) Chromatin fibrils in a pattern similar to a patchwork pattern. (**D**–**F**,**H**) Chromatin fibrils with spaces. (**A**) Stage VII. (**B**) Stage VIII. (**C**,**E**,**F**) Stage IX. (**D**) Antheridial filament fragment at stage IX. Some spermatids are not always arranged in the central section of the cell, but shift to one of the side walls. Nucleus with tighter spiral coils. (**G**–**I**) Stage X: (**I**) fragment of nucleus with the correct chromatin condensation. Longitudinal section of spermatids (**A**–**F**,**H**,**I**); transverse section of spermatid (**G**); av, autolytic vacuole; c, cytoplasm; cc, condensed chromatin; cw, cell wall; f, flagellum; mm, microtubular manchette; n, nucleus. Scale bar: 500 nm.

**Figure 5 cells-12-01268-f005:**
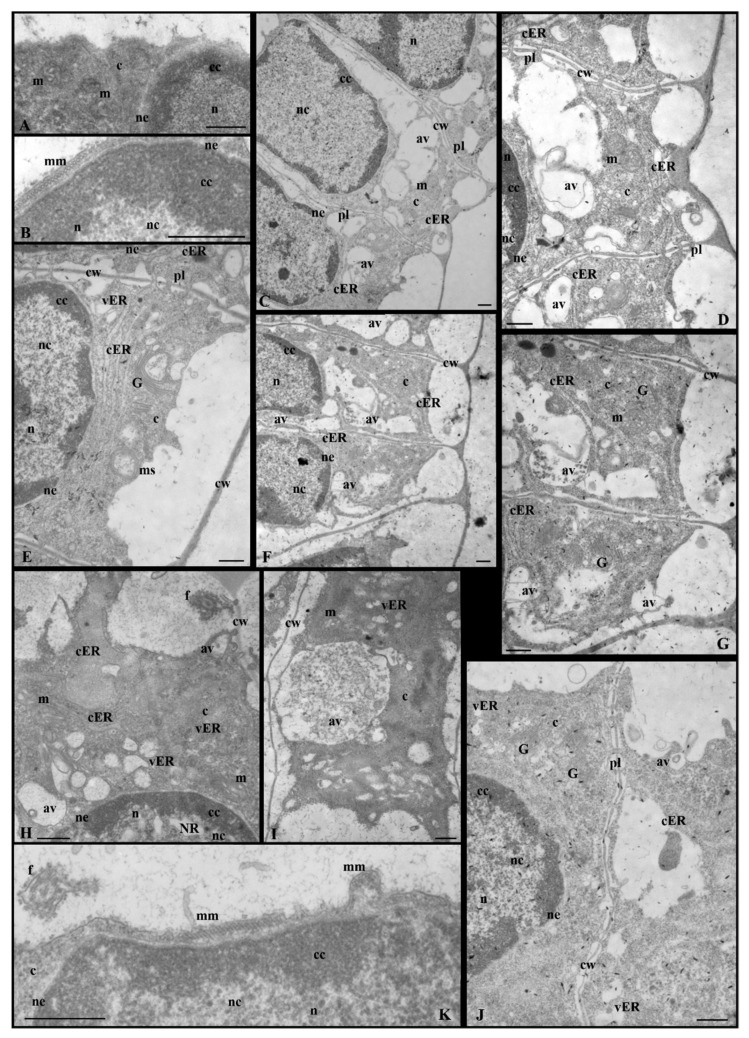
Ultrastructural changes in spermatid cytoplasm at stages III–IV following treatment with propyzamide; numerous Golgi apparatuses, ER cisternae, and vesicles; mitochondria with distended cristae, membranous structures, and autolytic vacuoles. (**A**,**B**) Control spermatid at stage IV, (**A**) mitochondria in the cytoplasm, and (**B**) fragment of microtubular manchette. (**C**,**D**,**E**,**J**) Stage III. (**F**,**G**,**H**,**I**,**K**) Stage IV. (**D**) Higher magnification of the cytoplasm area presented in (**C**). (**G**) Higher magnification of the spermatid presented in (**F**). (**H**) Extensive ER system and autolytic vacuoles. (**K**) Two fragments of the microtubular manchette with correctly and loosely arranged microtubules. Longitudinal section of spermatids; av, autolytic vacuole; c, cytoplasm; cc, condensed chromatin; cER, endoplasmic reticulum cisternae; cw, cell wall; f, flagellum; G, Golgi apparatus; m, mitochondrion; mm, microtubular manchette; ms, membranous structure; n, nucleus; nc, non-condensed chromatin; ne, nuclear envelope; NR, nuclear reticulum; pl, plasmodesmata; vER, endoplasmic reticulum vesicle. Scale bar: 500 nm.

**Figure 6 cells-12-01268-f006:**
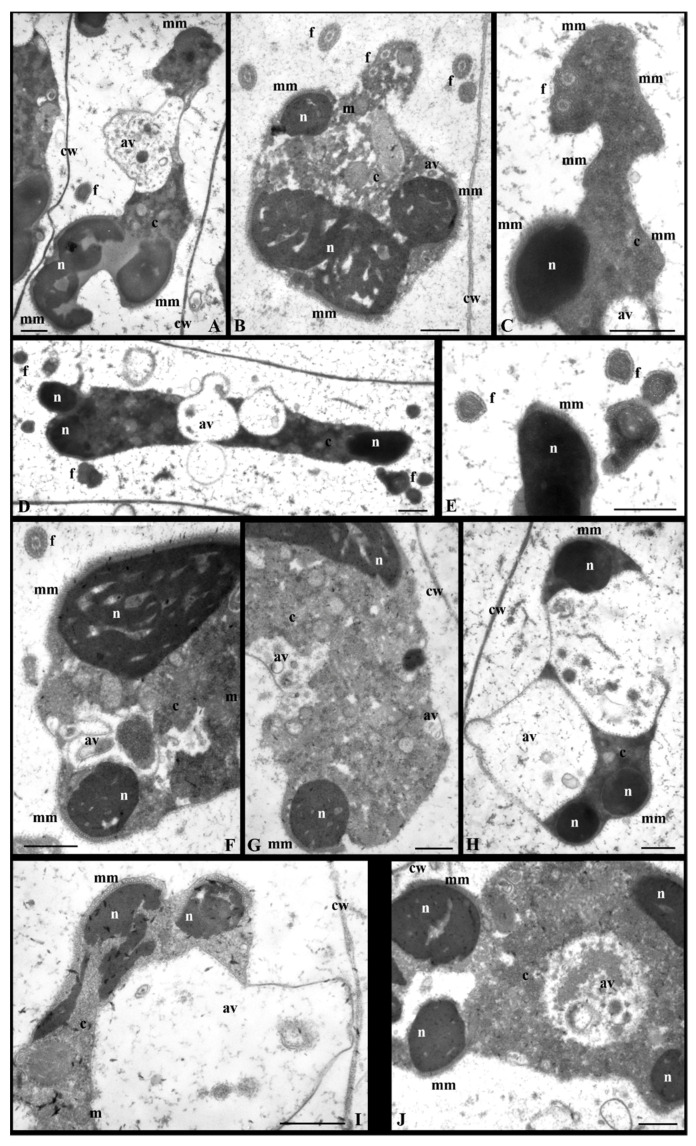
Ultrastructural changes in the spermatids’ cytoplasm during stages VIII–X of spermiogenesis following treatment with propyzamide. (**A**) Stage VIII. (**B**–**J**) Stages IX–X. Numerous large autolytic vacuoles filled with diverse materials (**A**,**D**,**F**,**H**–**J**). These vacuoles are located in the central sections of spermatids or in their peripheries. The spermatids with fragments of correct microtubular manchettes (**A**–**C**,**E**–**J**); some spermatids present a looser arrangement of microtubules in close proximity to the correct manchette (**C**) and flagella (**B**–**F**). Longitudinal section of spermatids; av, autolytic vacuole; c, cytoplasm; cw, cell wall; f, flagellum; m, mitochondrion; mm, microtubular manchette; n, nucleus. Scale bar: 500 nm.

**Table 1 cells-12-01268-t001:** The most important effects of propyzamide on the immunocytochemical and ultrastructural studies of *C. vulgaris* spermatids, separately presented for the nucleus and cytoplasm.

Control	Propyzamide
**Immunocytochemical studies** *β-tubulin immunosignals*	
- Stages I–X - Stages IV–VIII: clear reaction	- Stages I–III: weak/no signals - Stages IV–VI: weaker reaction - Stages VII–X: no signals
*Nucleus (DNA staining)*	
- Non-changed shape	- Distorted shape (some spermatids) - Diffused chromatin (some spermatids)
**Ultrastructural studies** ** *Nucleus* **	
Spermiogenesis stages I–V	
- Nuclear envelope: continuous - Condensed chromatin: near the nuclear envelope - Non-condensed chromatin: central part - Nuclear reticulum at stage V	- Nuclear envelope: invagination/fragmentation - Disturbances in chromatin structures (Condensed and non-condensed) - Looser system of non-condensed chromatin - Extensive nuclear reticulum already present at stage IV
Spermiogenesis stages VI–VIII	
- Correct chromatin condensation	- Disturbed chromatin condensation
- Stage VI: network system of chromatin	- Stage VI: non-network system of chromatin (large spaces)
Spermiogenesis stages IX–X	
- Strongly condensed chromatin - Spiral shape - Central part of the antheridial filament cell	- Disturbed chromatin condensation - Chromatin with bright electron spaces - Tighter spiral coils - Changed position within the antheridial filament cell
**Ultrastructural studies** *Cytoplasm*	
Spermiogenesis stages I–V	
- Numerous ER cisternae only at stage V - Linear microtubule manchette: stages IV–V	- Dilated ER cisternae/vesicles: stages III–V - Many extensive autolytic vacuoles - Fragmented microtubule manchette: stages IV–V
Spermiogenesis stages VI–VIII	
- Gradual reduction - Boundary, nucleus–cytoplasm: clearly visible - Correct microtubular manchette	- Lower degree of reduction - Boundary, nucleus–cytoplasm: less visible - Numerous ER vesicles - Disturbed microtubule manchette
Spermiogenesis stages IX–X	
- Significant reduction	- Abundant - Large autolytic vacuoles
** *Cell wall* **	
- Correct image	- Stratified, less visible (some spermatids)

## Data Availability

Not applicable.

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
