# Peer review of "Differentiation Disorders of Chara vulgaris Spermatids following Treatment with Propyzamide"

_cells, 2023, doi:10.3390/cells12091268_

Round 1

Reviewer 1 Report

The manuscript submitted by Agnieszka Wojtczak describes the effect of propyzamide, an inhibitor of ßtubulin on the differentiation of spermatids of the multicellular alga Chara vulgaris at the fine structural level. The propyzamide effect on Chara spermatids is compared with the action of propyzamide on spermatogenesis in animals. The results are interesting; however, extensive reorganization and shortening of the manuscript is required before it can be published.

1)     The number of figures must be reduced considerably; Fig. 2 and many electron micrographs are redundant. Less important images may be published as supplements. The details in the electron micrographs are often not discernible because of insufficient magnification and contrast. They should be visible in the printed version of the manuscript.

2)     The manuscript would benefit from a table describing briefly and concisely the most important effects of propyzamide. Alternatively or in addition, a schematic illustration would help the reader to get a better impression what happens.

3)     The introduction and discussion must be shortened too. Instead of “herbicides” the term “inhibitor” should be used because this manuscript deals with basic science not with applied science. Discussion should focus on inhibitors of tubulin/microtubules. Especially reference 18 should be more considered since it deals with the effect of another microtubule inhibitor on Chara spermatogenesis. If other inhibitors are mentioned, please give information about their action and target (e. g. atrazine, roundup). Please use scientific names not brand names (glyphosate instead of roundup).

Other comments:

L  60: microtubules are not involved in ctoplasmic streaming in characean algae

L 114: please give details of DMSO concentrations

L 127: which embedding medium?

L 257: what does openwork mean?

L 444: please explain – are these cells dead? - if there is no cell membrane?

L 427: what is an electronogram? Perhaps replace by electron micrograph?

Fig 1: Please explain what is stained here, is it the manchette? Why are flagella not stained with this antibody?

Fig 3 legend: C, D are not mentioned

Figs 4, 7, 8, 10: Please separate images in figures by equally thick lines, not by gaps

Fig 8 legend: “cytoplasm” – everything within and including the cell membrane is cytoplasm, replace by cytosol?

Please check carefully for typing errors and grammatical mistakes (e.g. lines 95,107,128,129,135,409,521….)

Author Response

Replies to Reviewer 1 comments:

Author would like to thank the Reviewer for valuable comments which helped to improve the paper.

The changes in the manuscript were highlighted with the use "Track Changes" function in Microsoft Word. Number of lines is according to earlier submitted manuscript.

The manuscript has been extensively revised in English by the Language Editing Services.

Reply 1

According to the suggestion the number of figures in the main text was reduced. Other figures (Figures 2, 5, 6, 9) are included as supplements. Figure numbers were changed (details in the table below).

Previous figure

number

Figure number

in revised version

Fig. S1 (new)

Fig. 1

Fig. 1

Fig. 2

Fig. S2

Fig. 3

Fig. 2

Fig. 4

Fig. 3

Fig. 5

Fig. S3

Fig. 6

Fig. S4

Fig. 7

Fig. 4

Fig. 8

Fig. 5

Fig. 9

Fig. S5

Fig. 10

Fig. 6

The contrast in the images has been improved. Because the photos were taken on a microscope that does not have a digital image saving system, there were no possibility to evaluate them immediately.

Reply 2

Table 1. describing the most important effects of propyzamide, separately for nucleus and cytoplasm, was added to the manuscript.

Reply 3

The introduction and discussion were shortened. Some references were deleted ([68,75,76,78]) and some references were added [current number 81-84] as suggested by another Reviewer.

The term “herbicide” was exchanged into “inhibitor”. Name “oryzalin” was exchanged into “dinitroaniline”. Information about the action of nocodazole (it interferes with microtubule polymerization) was added.

Other comments:

L  60: microtubules are not involved in ctoplasmic streaming in characean algae 

Reply L  60: “… the involvement of microtubules in cytoplasmic streaming “ – was deleted

L 114: please give details of DMSO concentrations

Reply L 114: detail of DMSO concentrations – “ …dimethyl sulfoxide (DMSO) (Sigma D-4540) in the 0.1% ratio …” was added

L 127: which embedding medium?

Reply L 127: details of embedding medium were added, „Following the staining procedure, preparations were embedded in a PBS/glycerol mixture (9:1) with 2.3% DABCO (1,4-diazabicyclo-[2,2,2] octane, Sigma).”

L 257: what does openwork mean?

Reply L 257: I used “openwork” as area with spaces and holes.

L 444: please explain – are these cells dead? - if there is no cell membrane?

Reply L 444: sentence “At these stages, the cytoplasm was somewhat diffused, because the damaged plasmalemma did not limit its area.” was exchanged into “At these stages, the cytoplasm was somewhat diffused, because the plasmalemma was presumably partially damaged.”

L 427: what is an electronogram? Perhaps replace by electron micrograph?

Reply L 427: “electronogram” was exchanged into “electron micrograph”

Fig. 1 please explain;

Reply – Paraffin sections were used as the material for immunocytochemical studies. Not on all pictures numerous cross-sections of flagella are visible.

Immunocytochemical foci visible on the cytoplasm area – this is a manchette and near the cytoplasm - flagella.

Fig 3 legend:

Reply - Fig 3 legend: has been supplemented

“… The control spermatids – (A) at all presented stages and (B) at IX-X stages; spermatids after propyzamide treatment – the remaining images; …” was exchanged into “… Control spermatids – (A) at all presented stages and (B) at IX-X stages; spermatids following propyzamide treatment – (B) at III-IV, V, VI and VIII; (C) at III-IV, V and IX-X; and (D) at III-IV stages; …”

Figs 4, 7, 8, 10:

Reply - Images in Figures 4, 7, 8, 10 were separated by thick lines.

Fig 8 legend: 

Reply - I think that legend “cytoplasm” will be correct. I mean cytoplasm as cell area containing the cytosol and organelles.

Reviewer 2 Report

The author has claimed that propyzamide had an effect on Chara spermiogenesis, and it is shown for the first time in this manuscript.

Several comments regarding the manuscript, especially in the method section:

- Line 112, Please mention why the water of the natural environment was used? 

- Line 113, Please mention the ratio used.

- Line 114, Why did the control only contained DMSO, and not contain the water from the water environment.

- Line 110, Line 116, What were the positive controls in these experiments?

- Line 145, Please mention what the green and blue colour shows.

- Line 145, Please mention what I-VIII, and X, refers to.

- Line 150, "weak immunosignals were measured visible", how were these signals measured? Please show the graphs/spectras.

Author Response

Replies to Reviewer 2 comments:

Author would like to thank the Reviewer for valuable comments which helped to improve the paper.

The changes in the manuscript were highlighted with the use "Track Changes" function in Microsoft Word. Number of lines is according to earlier submitted manuscript.

The manuscript has been extensively revised in English by the Language Editing Services.

- Line 112

Reply – information “The solutions were then prepared with water collected from the natural environment in order to not affect the conditions for the algae’s growth.” was added.      

In our previous works the solutions of all inhibitor were prepare in the mentioned way.

- Line 113,

Reply - details of DMSO concentrations – “ …dimethyl sulfoxide (DMSO) (Sigma D-4540) in the 0.1% ratio …” were added

- Line 114,

Reply - In earlier Chara studies control contained water from the natural environment. Since no differences were observed, this control was not used in further experiments.

- Line 110, Line 116

Reply - There were no positive controls in this experiment. This is the first study on the effects of propyzamide on Chara. Inhibitors of microtubules have not been used before.

- Line 145 (green and blue color)

Reply - Fig 1 legend: has been supplemented

Figure 1. Immunocytochemical localization of b-tubulin protein in antheridial filament cells obtained from the selected stages of C. vulgaris spermiogenesis. (A,B) the control stages, clear reaction – stages IV-VIII, (C,D) following 48 h treatment with propyzamide, weaker reaction – stages IV-VI. Secondary antibodies conjugated with FITC (A,C), positive fluorescence is marked with green foci. Nuclei stained with DAPI (B,D) and colored in blue. Scale bar, 10 µm.”

- Line 145  (I-VIII, and X)

Reply - Fig. 1 has been supplemented, the designation of spermiogenesis stages was added.

These C. vulgaris spermiogenesis stages were distinguished on the basis of previous light and electron microscopy observations (Kwiatkowska and Popłońska 2002 [42]) and characteristic ultrastructural changes.

- Line 150

Reply - A graph (Fig. S1) concerning  fluorescence intensity measurements  was added.

Text was added, line 129 - “The fluorescence intensity in spermatids was measured from selected spermiogenesis stages (presented in Figure 1); for the control and propyzamide-treated variant, using ImageJ (Figure S1). The average value and standard deviation (means ± SD) were calculated with the use of Microsoft Excel 2000.”

Because the figure numbers have changed, I have attached a table with the changes below.

Previous figure

number

Figure number

in the revised version

Fig. S1 (new)

Fig. 1

Fig. 1

Fig. 2

Fig. S2

Fig. 3

Fig. 2

Fig. 4

Fig. 3

Fig. 5

Fig. S3

Fig. 6

Fig. S4

Fig. 7

Fig. 4

Fig. 8

Fig. 5

Fig. 9

Fig. S5

Fig. 10

Fig. 6

Reviewer 3 Report

1- The authors could refer to some more most recent references from the Year 2022

2- Besides, the Results and Discussion (sub-titles) should correspond with the Materials and Methods (sub-titles). The authors could look into this and revise/amend accordingly.   

 3- Conclusions: Should summarize the significant findings of ALL THE MAJOR STUDIES, and thus needs to be extended further within the scopes.

4- Ideally, the figure captions should be informative and representative. Currently, most of the captions are just the name of the figure(s) and thus require further extension within the limit.

Author Response

Replies to Reviewer 3 comments:

Author would like to thank the Reviewer for valuable comments which helped to improve the paper.

The changes in the manuscript were highlighted with the use "Track Changes" function in Microsoft Word. Number of lines is according to earlier submitted manuscript.

The manuscript has been extensively revised in English by the Language Editing Services.

Reply 1 - Four new references [current number 81-84] were added. The difficulty is that the literature does not always concern both spermiogenesis and microtubules.

sentence was added – L 533, “Studies conducted on mouse spermatogenesis using a chemotherapeutic agent, vincristine, which inhibits the functioning of spindle microtubules, are important for the cancer treatment  regarding testicular tissue in human beings [81,82].”

sentences was added – L 536, “Recently, a tubulin inhibitor Todalam, with a specific mechanism of action that inhibits mammalian cell growth was designed [83]. The research conducted on the participation of microtubule-associated proteins in the regulation of microtubule dynamics and in plant development is being conducted, at present. Furthermore, it has been shown in the research that MICROTUBULE ORGANIZATION 1 (MOR1) protein is essential to mitosis and cytokinesis processes in Arabidopsis. A seedling analysis showed that MOR1 mutants are more sensitive to propyzamide treatment [84].”

Reply 2 - In the Discussion sub-titles, Immunocytochemical studies and Ultrastructural studies were added. The final part of the Discussion, however, refers to the generally discussed issues.

Reply 3 - As a summary, Table 1 describing the most important effects of propyzamide, separately for nucleus and cytoplasm, was added to the manuscript.

Reply 4 - Figure captions were slightly supplemented. Additional information was not included in the figure caption to avoid repetition from the main text.

Because the figure numbers have changed, I have attached a table with the changes below

Previous figure

number

Figure number

in the revised version

Fig. S1 (new)

Fig. 1

Fig. 1

Fig. 2

Fig. S2

Fig. 3

Fig. 2

Fig. 4

Fig. 3

Fig. 5

Fig. S3

Fig. 6

Fig. S4

Fig. 7

Fig. 4

Fig. 8

Fig. 5

Fig. 9

Fig. S5

Fig. 10

Fig. 6

Round 2

Reviewer 1 Report

Line 60: please add

... for the process of actin-dependent cytoplasmic streaming....

Lines between images in the revised version are now black or white. Please change to a uniform style and, if possible, use equally thick lines.